# Assessment of Methylation in Selected *ADAMTS* Family Genes in Non-Small-Cell Lung Cancer

**DOI:** 10.3390/ijms26030934

**Published:** 2025-01-23

**Authors:** Dagmara Szmajda-Krygier, Zuzanna Nocoń, Jacek Pietrzak, Adrian Krygier, Ewa Balcerczak

**Affiliations:** 1Department of Pharmaceutical Biochemistry and Molecular Diagnostics, Medical University of Lodz, Muszynskiego 1, 90-151 Lodz, Poland; 2BRaIn Laboratories, Medical University of Lodz, Czechoslowacka 4, 92-216 Lodz, Poland

**Keywords:** *ADAMTS*, epigenetics, LUAD, lung cancer, LUSC, methylation

## Abstract

Alterations in the methylation of genetic material can influence carcinogenesis by the downregulation or overexpression of *ADAMTS* (a disintegrin-like and metalloprotease with thrombospondin motifs) protease genes. Through their proteolytic activity, these enzymes are also capable of promoting angiogenesis. Consequently, ADAMTS proteases can either facilitate or inhibit cancer progression. This study aimed to evaluate the methylation levels of the *ADAMTS6*, *ADAMTS9*, and *ADAMTS12* genes in non-small-cell lung cancer (NSCLC) using data from bioinformatics databases. The focus was on differences between lung adenocarcinoma (LUAD) and lung squamous-cell carcinoma (LUSC) subtypes and their impact on patient overall survival (OS). *ADAMTS6* gene expression is significantly reduced in LUSC, and analysis of *ADAMTS9* gene expression showed a significantly reduced gene transcript level in LUAD and LUSC, while both NSCLC subtypes demonstrated *ADAMTS12* upregulation. In LUSC, significantly elevated promoter methylation was found in all of the aforementioned genes, while in LUAD, higher promoter methylation was observed only for *ADAMTS9* and *ADAMTS12*. The differential methylation region (DMR) pattern demonstrated by *ADAMTS6*, *ADAMTS9*, and *ADAMTS12* is a useful tool for distinguishing normal from cancer cells. The areas under the curve (AUCs) ranged from 0.86 to 0.99 for both LUAD and LUSC subtypes. The methylation level of different CpG sites among selected *ADAMTS* members is related to patient survival, suggesting it may have value as a prognostic marker. The methylation degree of promoter regions in genes encoding ADAMTS family proteins could significantly influence LUSC and LUAD. Increased promoter methylation could also reduce certain gene expression, contributing to cancer progression. The expression levels and specific DMRs of *ADAMTS* genes may serve as prognostic markers correlating with patient OS. Assessing *ADAMTS* gene methylation could become a diagnostic tool for differentiating NSCLC subtypes and potentially guide therapeutic strategies. Further research is needed to fully understand the activity and mechanisms of ADAMTS family proteins.

## 1. Introduction

Lung cancer poses a particularly serious threat to human life and health. This may be attributed to the non-specific symptoms of the disease, or their absence, late detection, and difficulties in selecting appropriate treatment. Most worryingly, its incidence is currently rising, particularly among women [1].

A critical factor in the development of lung cancer is DNA methylation. Aberrations in the attachment of methyl groups to genetic material predominantly result in the silencing of tumor suppressor gene promoters [2]. Such methylation frequently occurs in CpG islands, affecting either the transcription start site or the enhancer regions [3]. The addition of a methyl group to the 5-carbon of the cytosine ring within CpG islands epigenetically influences gene expression. This process can be triggered by aging, disease progression, or environmental factors. Methylation of the promoter region inhibits DNA transcription by obstructing the binding sites for RNA polymerase or other transcription factors [4]. Numerous studies confirm that increased frequency and levels of hypermethylation, or single gene methylation events, are associated with an accelerated progression of lung cancer [2].

Matrix metalloproteinases are often associated with increased tumor progression. It is assumed that they support the migration of cancer cells and can facilitate tumor growth by contributing to the degradation of elements of the extracellular matrix [3]. An analysis of topographic data relating to mutations and gene expression in cancer patients documented the presence of as many as 38 genes that may be associated with the occurrence of cancer [5]. It is assumed that some members of the ADAMTS (a disintegrin-like and metalloprotease with thrombospondin motifs) protein family may act as tumor suppressors or oncogenes [3]. A study conducted on mice showed that a deletion in the *ADAMTS12* gene increases the likelihood of lung cancer by up to five times [6]. Reducing the expression of the *ADAMTS12* gene in lung cells affected by cancer resulted in increased cell proliferation and invasiveness [7], indicating that *ADAMTS12* may act as a tumor suppressor gene [6].

For the *ADAMTS9* gene, it was shown that it acts as a tumor suppressor in gastric cancer through AKT/mTOR signaling inhibition [8]. In esophageal and nasopharyngeal carcinoma, the *ADAMTS9* expression level was low due to promoter hypermethylation [9]. What is more, in patients with gastric cancer, higher *ADAMTS9* methylation was significantly associated with shortened survival, indicating its value as a prognostic factor [8].

Other studies have shown that *ADAMTS6* may be associated with the occurrence of NSCLC (non-small-cell lung cancer) by affecting the regulation of AGR2 expression. High levels of the AGR2 protein are seen in drug-resistant NSCLC. The enzyme is likely involved in the proliferation and migration of cancer cells. The interaction between the expressions of both proteins operates as a negative feedback loop: elevated levels of ADAMTS6 protein suppress AGR2 expression, thereby diminishing angiogenesis [10].

*TP53* is a suppressor gene whose mutation status is connected to prognosis and may be a factor influencing responses to chemotherapy and radiation [11]. It is said to be altered in more than 50% of all lung tumors and is considered an undruggable target [12]. Patients harboring mutant *TP53* have worse overall survival and are more prone to chemotherapy resistance [12,13]. That is why we included, in a subsequent analysis, the status of *TP53* mutation as a factor influencing NSCLC patients´ survival.

In summary, alterations in the methylation of genetic material can contribute to carcinogenesis by the downregulation or overexpression of ADAMTS protease genes [3]. Through their proteolytic activity, these enzymes are also capable of promoting angiogenesis. Consequently, ADAMTS proteases can either facilitate or inhibit cancer progression [6].

The aim of the present study was to evaluate the degree of methylation of the *ADAMTS6*, *ADAMTS9*, and *ADAMTS12* genes in NSCLC using data from bioinformatics databases, with a particular focus on differences between LUAD (lung adenocarcinoma) and LUSC (lung squamous-cell carcinoma) subtypes and their impact on patient OS (overall survival).

## 2. Results

### 2.1. ADAMTS Expression and Methylation Analysis in LUAD and LUSC

In the first stage of the analysis, the mRNA expression of the *ADAMTS6*, *ADAMTS9*, and *ADAMTS12* genes was assessed in lung cancer, and the obtained results were compared with data from adjacent normal tissues. The analysis was performed using the DiffExp module available on the TIMER (Tumor Immune Estimation Resource) platform based on the TCGA (The Cancer Genome Atlas) RNA-seq dataset.

The *ADAMTS6* gene was found to demonstrate significantly lower expression in lung squamous-cell carcinoma than in normal cells (*p* < 0.05; Figure 1A). However, in lung adenocarcinoma, the differences between gene expression in cancer-free cells and cancer tissue were not significant.

Reduced expression of the *ADAMTS9* gene was found for LUAD and LUSC (*p* < 0.001; Figure 1B). While for *ADAMTS12*, overexpression was observed in LUAD (*p* < 0.001) and LUSC (*p* < 0.05) (Figure 1C).

Different transcript levels of *ADAMTS6*, *ADAMTS9*, and *ADAMTS12* are observed in various types of cancer, which points to the problematic part played by these proteases in the pathogenesis of proliferative diseases. The results also prove the existence of a multi-faceted correlation between the expression of the indicated genes and the pathogenesis of many types of cancer.

In the next step, the degree of methylation was assessed in the promoter region of three selected genes of the ADAMTS protein family using UALCAN (The University of Alabama at Birmingham Cancer Data Analysis Portal) analytical tools (Figure 2). The findings indicate significantly higher *ADAMTS6*, *ADAMTS9*, and *ADAMTS12* promoter methylation in the tissue affected by cancer compared with normal tissue in patients with squamous-cell lung cancer (*p* < 0.001). However, in patients with lung adenocarcinoma, no significant difference in methylation was observed for *ADAMTS6* between tissue types (Figure 2A), while methylation was significantly higher in cancer tissue compared with normal lung tissue for *ADAMTS9* (*p* < 0.001) and *ADAMTS12* (*p* < 0.001).

### 2.2. The Methylation Level in Relation to Various Clinicopathological and Demographic Factors

The next step was to compare the methylation levels of the *ADAMTS6*, *ADAMTS9*, and *ADAMS12* promoter regions with various clinicopathological and demographic factors among patients with LUAD and LUSC using the data available in the UALCAN database. The factors discussed included age, sex, ethnicity, smoking status, tumor advancement, the presence of mutations in the *TP53* gene, and the number of metastases to nearby lymph nodes.

#### 2.2.1. ADAMTS6

Statistically significant differences in promoter methylation were observed for the *ADAMTS6* gene with regard to the UALCAN data, particularly among patients with squamous-cell lung cancer. Only slight differences in methylation were observed among patients with lung adenocarcinoma, and these were primarily associated with cancer stage and smoking status.

No significant correlations were found in patients with LUAD with regard to sex, age, ethnicity, the presence of *TP53* mutations, or the number of metastases to local lymph nodes (*p* > 0.05; Figure 3 and Figure 4).

#### 2.2.2. ADAMTS9

No significant differences in *ADAMTS9* gene methylation were noted among patients with adenocarcinoma and squamous-cell lung cancer with regard to sex or smoking status (*p* > 0.05; Figure 5A,D).

In patients with LUSC, the only significant changes in *ADAMTS9* promoter methylation were observed with regard to ethnicity and the presence of mutations in the *TP53* gene (Figure 5B and Figure 6A, respectively). Higher levels of gene promoter methylation were found in Asian patients compared with Caucasians (*p* = 0.020) and African Americans (*p* = 0.022; Figure 5B). However, this result may be due to the small size of the Asian subgroup (n = 7). No significant differences in lung squamous-cell carcinoma were noted between Caucasian and Black patients (*p* > 0.05).

#### 2.2.3. ADAMTS12

In neither lung squamous-cell carcinoma nor lung adenocarcinoma patients, no statistically significant differences were observed in the level of *ADAMTS12* gene promoter methylation between male and female patients (*p* > 0.05; Figure 7A). Additionally, in LUAD and LUSC, no significant differences in *ADAMTS12* methylation were found between patients with and without the *TP53* mutation (*p* > 0.05; Figure 8A).

In the group of patients with lung adenocarcinoma, significant differences in *ADAMTS12* promoter methylation were noted with regard to cigarette smoking and ethnic origin. A higher level of gene methylation was observed in Caucasian patients compared with African American patients (*p* = 0.026). No other significant differences were found between other subgroups (*p* > 0.05; Figure 7A). No significant differences were found among patients with lung squamous-cell carcinoma (*p* > 0.05; Figure 7B).

### 2.3. Clinical and Diagnostic Value of ADAMTS6, ADAMTS9, and ADAMTS12 Promoter Methylation Patterns

The MethMarkerDB tool analysis, using the ‘MethMarker gene search’ module, revealed that in the LC samples, *ADAMTS6*, *ADAMTS9*, and *ADAMTS12* had several differentially methylated regions (DMRs) compared with the normal samples (Figure 9). In all three genes, the promoter region was hypermethylated (*p* < 0.001; beta-value difference: 0.508, 0.321, and 0.377, respectively), while other regions were both hypo- and hypermethylated. The greatest number of differences between normal and lung cancer samples was observed for the *ADAMTS12* gene (n = 110); these changes were considerably more frequent than in *ADAMTS6* and *ADAMTS9* (DMRs: n = 7 and n = 10, respectively).

The diagnostic value of the DMRs of the selected genes was assessed in the LUAD and LUSC samples using the ‘MethMarker Evaluation’ module. It was found that the distinct methylation pattern among *ADAMTS* members effectively distinguished lung cancer samples from normal samples (Figure 10). The highest AUC value was exhibited by the *ADAMTS12* gene in both the LUAD and LUSC cohorts (Figure 10E, 0.989 and Figure 10F, 0.999, respectively).

### 2.4. Survival Analysis Based on Methylation Levels of ADAMTS6, ADAMTS9, and ADAMTS12 Genes

The influence of individual CpG sites on patient survival was assessed with the MethSurv database. The analysis revealed that for several CpG sites, the degree of methylation corresponds with changes in LC patient survival.

In the LUAD cohort, for *ADAMTS6*, higher methylation at the cg21878650 and cg24166172 sites was associated with a shorter OS (*p*= 0.022 and *p* = 0.021, respectively; Figure 11A,B); in contrast, for the *ADAMTS9* gene, lower methylation at cg00508543 and cg03427905 corresponded with a shorter OS (*p* = 0.016 and *p* = 0.049; Figure 11C,D), and lower methylation at cg22177868 with a longer OS (*p* = 0.005; Figure 11E). For *ADAMTS12*, high methylation at cg23359363 correlated with a shortened OS in the LUAD cohort (*p* = 0.049; Figure 11F).

In the LUSC cohort, for *ADAMTS6*, a shorter OS was associated with higher methylation at cg00299603 and cg19927028 (*p* = 0.039 and *p* = 0.005, respectively; Figure 12A,C) and lower methylation at cg14793753 and cg26688155 (*p* = 0.016 and *p* = 0.037; Figure 12B,D). Similar to the LUAD cohort, in the *ADAMTS9* gene, lower methylation at cg07891473 and cg11427510 corresponded with a shorter OS (*p* = 0.034 and *p* = 0.021; Figure 12E,F) and lower methylation at cg25157607 with a longer OS (*p* = 0.042; Figure 12G). Lastly, for *ADAMTS12*, higher methylation at cg04578894 and cg06448603 corresponded significantly with a shorter OS (*p* = 0.028 and *p* = 0.041, respectively; Figure 12H,I), whereas high methylation at cg10594543 and cg23359363 was connected with better survival (*p* = 0.021 and *p* = 0.039; Figure 12J,K).

## 3. Discussion

ADAMTS proteins are zinc-dependent metalloproteinases secreted into the extracellular matrix and are found in almost all human tissues. Their widespread presence correlates with their diverse roles and the potential for numerous disorders. ADAMTS6, ADAMTS9, and ADAMTS12 possess proteolytic activity, which plays a key role in extracellular matrix remodeling [14]. They exhibit diverse substrate preferences and tissue-specific expression and, thus, varied proteolytic activity. The known functions of *ADAMTS6*, *ADAMTS9*, and *ADAMTS12* include the regulation of cardiovascular processes, immune responses, and cartilage and bone tissue reorganization. They can act as tumor suppressors or oncogenes in various cancers. Their altered expression may contribute to pathological processes and cancer progression [15,16,17].

This study aimed to determine the methylation levels of the *ADAMTS6*, *ADAMTS9*, and *ADAMTS12* genes in NSCLC using data from bioinformatics databases, with a particular focus on differences between LUAD and LUSC subtypes and their impact on patient OS. As no previous studies have assessed the expression and methylation levels of these genes in lung cancer, no direct comparison with other research is possible.

The gene expression data available in the databases indicated differences between tumors and normal tissue with regard to the regulation of *ADAMTS6*, *ADAMTS9*, and *ADAMTS12*, which may be related to their different roles in cancer pathogenesis. *ADAMTS6* gene expression is significantly reduced in LUSC compared with normal tissue but not in LUAD; this suggests that *ADAMTS6* gene transcript regulation may differ between NSCLC histological subtypes. Reduced *ADAMTS6* gene transcript levels have also been observed in bladder urothelial carcinoma (BLCA), kidney chromophobe (KICH), kidney renal clear-cell carcinoma (KIRC), kidney renal papillary-cell carcinoma (KIRP), skin cutaneous melanoma (SKCM), and uterine corpus endometrial carcinoma (UCEC). Increased gene expression compared with adjacent normal tissues has been demonstrated in patients with breast invasive carcinoma (BRCA), invasive cholangiocarcinoma (CHOL), colon adenocarcinoma (COAD), esophageal carcinoma (ESCA), head and neck squamous-cell carcinoma (HNSC), liver hepatocellular carcinoma (LIHC), rectal adenocarcinoma (READ), and stomach adenocarcinoma (STAD). A previous study on the effect of *AGR2* overexpression on drug resistance also found *ADAMTS6* expression to be reduced in NSCLC; however, the experiment did not specify the histological subtype of the studied tumor. An analysis of *ADAMTS9* gene expression showed a significantly reduced gene transcript level in LUAD and LUSC compared with normal tissue [10]. Similar *ADAMTS9* gene expression was reported in a LUAD bioinformatics study by Liu et al. [18] based on Gene Expression Omnibus (GEO) data. No similar studies on the expression of this gene in LUSC have been published to date. Additionally, for most charted tumor types, *ADAMTS9* gene expression was significantly higher in normal tissues than in the associated tumor samples, and overexpression of the *ADAMTS9* gene was observed only in thyroid carcinoma (THCA), CHOL, COAD, KIRP, LIHC, and STAD. *ADAMTS12* expression was significantly higher in both LUAD and LUSC tissue compared with non-cancerous tissue. The *ADAMTS12* gene was also upregulated in various cancer types, such as CHOL, COAD, ESCA, HNSC, READ, and STAD. In contrast, no such difference was noted in an integrated pan-cancer analysis of the *ADAMTS12* gene, directed mainly against pancreatic adenocarcinoma [19]. *ADAMTS12* expression was found to be lower than in the corresponding normal tissues in prostate adenocarcinoma (PRAD), BRCA, KICH, KIRP, SKCM, and UCAC.

The current literature suggests that the *ADAMTS* family plays an ambiguous role in carcinogenesis, with its regulation varying depending on the type of cancer. Both *ADAMTS6* and *ADAMTS9* gene expression is reduced in lung cancer, which may potentially be associated with the role of the genes as tumor suppressors. In contrast, the fact that *ADAMTS12* is overexpressed may suggest that it has oncogenic potential. However, further in-depth research is needed to fully understand the function of the genes in lung cancer.

The next part of the analysis concerned the differences in the degree of *ADAMTS6*, *ADAMTS9*, and *ADAMTS12* promoter methylation between LUAD and LUSC. It was found that promoter methylation was significantly elevated in all of the tested genes in LUSC; however, in LUAD, a higher degree of promoter methylation was observed only for *ADAMTS9* and *ADAMTS12*. Little is known so far about the relationship between promoter methylation of ADAMTS protein family genes and carcinogenesis, especially in the case of *ADAMTS6*.

Like *ADAMTS6*, *ADAMTS16* encodes an orphan enzyme. The gene demonstrated similar general methylation profiles and methylation changes in the three tested cancer samples: lung cancer, oral squamous-cell carcinoma, and colorectal cancer. In all cases, six CpG islands were hypermethylated compared with non-cancerous tissue. The change reduced the level of the *ADAMTS16* gene transcript, thus increasing the risk of colorectal cancer [3].

In 2019, Zhang et al. [20] also noted an association between increased methylation of CpG islands in the *ADAMTS18* gene promoter and decreased expression compared with adjacent normal tissues. Similarly, *ADAMTS9* gene expression was lowered in LUAD and LUSC, which correlated with increased promoter methylation levels.

From the available publications on NSCLC, a study by Choi et al. [21] noted transcriptional inactivation of the *ADAMTS1* gene due to aberrant methylation. This association has also been demonstrated for *ADAMTS9* in gastric cancer [8], breast cancer [22], and multiple myeloma cells [23]. These findings also indicate that ectopic *ADAMTS9* expression in cancers results in the inhibition of cancer cell proliferation, a reduction in VEGF expression, and the induction of cancer cell apoptosis. This may suggest that the gene has potential anti-angiogenic activity and an inhibitory effect on the carcinogenesis process. Our present findings indicate increased *ADAMTS12* gene expression in LUAD and LUSC, with significantly increased promoter methylation compared with normal tissue.

In contrast, Daniunaite et al. [24] reported high methylation of the *ADAMTS12* gene promoter in prostate cancer tissue compared with normal tissue, with reduced expression of mRNA in cancer cells. Ambiguous results were obtained by Moncada-Pazos et al. [25] in an analysis of *ADAMTS12* gene hypermethylation in colorectal cancer. Promoter methylation was found to be elevated in the tumor material compared with normal tissue; in addition, RT-qPCR amplification showed no significant *ADAMTS12* expression in selected microsatellite-stabilized colon cancer cell lines. It, hence, appears that promoter methylation of *ADAMTS12* could lead to its silencing. In contrast, an additional RT-qPCR analysis showed significantly higher levels of *ADAMTS12* expression in cancer samples used for the methylation study than in normal tissues. A similar result was obtained using commercially available cDNA samples of colon cancer and normal tissue. Finally, the stromal cells surrounding the colon cancer cells were found to be responsible for the increased expression of *ADAMTS12*. Researchers have suggested that this may be a compensatory mechanism for the reduced gene transcript in cancer cells, which is supposed to have a protective function. Therefore, the expression of *ADAMTS12* in LUAD and LUSC may potentially be regulated by promoter methylation; however, it is also possible that other additional mechanisms at the transcription level (e.g., microRNA), as well as post-transcription and post-translational mechanisms, including alternative splicing and activation via furin [7], may also be involved. It is also possible that, in the case of the studied NSCLC subtypes, the gene itself may be involved in the process of carcinogenesis and may promote tumor development. Further in-depth research in this direction is needed.

This study also analyzed the effects of various clinical and pathological factors on the methylation of the promoter regions of the discussed genes. It was found that sex is not a factor influencing the methylation of the promoter region in patients with LUSC and LUAD. Similar results were obtained in a study on multiple myeloma [23] and gastric cancer [8], where no significant differences in *ADAMTS9* gene methylation were found depending on sex. An analysis examining the dependence of gene methylation on ethnicity showed that African American patients with LUSC are characterized by an increased level of methylation in the promoter region of the *ADAMTS6* gene compared with Caucasian and Asian patients. Increased methylation of the *ADAMTS9* promoter region was observed in Asian patients with LUSC compared with the African American and Caucasian groups. Caucasian patients with LUAD demonstrate an increased degree of *ADAMTS12* and *ADAMTS9* promoter methylation compared with African Americans. It seems that the *ADAMTS* genes vary depending on ethnicity; this may also be a factor in the discrepancies noted in the previously obtained analyses.

Our present findings indicate that patients with LUAD over 80 years of age show a higher degree of *ADAMTS9* gene promoter methylation than those aged 41–80 years. However, age was not found to have any significant influence on promoter methylation in previous research on multiple myeloma [23], breast cancer [22], or gastric cancer [8]. In the present study, it was also found that LUSC patients aged 80–100 years have a lower degree of methylation in the promoter region of the *ADAMTS12* gene than those aged 41–60 years.

Both NSCLC subtype groups indicated that cigarette smoking has an effect on *ADAMTS6* gene methylation levels. Non-smokers with LUAD showed lower promoter methylation than ex-smokers and lower *ADAMTS12* gene methylation compared with those who quit smoking less than 15 years previously. Active smokers with LUAD showed lower levels of *ADAMTS6* gene promoter methylation than those who had quit within the previous 15 years. In the LUSC group, smokers who had quit within 15 years demonstrated higher *ADAMTS6* promoter methylation than long-term non-smokers. The results suggest that smoking may have an impact on the methylation of the promoter regions of the highlighted genes.

Among patients with LUAD, those with stage 3 showed significantly reduced *ADAMTS9* gene promoter methylation than those in less-advanced stages of the disease. It is possible that with the advancement of the tumor, the tendency to methylate this particular promoter region decreases. No such differences in *ADAMTS9* methylation in relation to tumor stage were observed in multiple myeloma [23], gastric cancer [8], or breast cancer [22]. Reduced *ADAMTS6* methylation was also demonstrated in patients with stage 4 LUAD compared with those in stage 2. In the case of LUSC, higher *ADAMTS12* promoter methylation was associated with stage 2 cancer than stage 1.

Patients with LUAD with metastases in 4–9 lymph nodes demonstrated significantly lower *ADAMTS9* gene methylation than those with fewer or no metastases. However, in patients with LUSC, higher *ADAMTS12* promoter methylation was observed in patients with metastases in 1–3 lymph nodes compared with those without metastases.

Finally, LUSC patients with a normal variant of *TP53* show a higher degree of *ADAMTS9* methylation than those with a mutated form. No similar influence has been demonstrated for the other tested genes or for LUAD, nor in previously published scientific articles.

One potential prognostic and diagnostic feature of the tested *ADAMTS* gene family members merits further consideration. The bioinformatic analyses found the presence of a distinct DMR pattern among *ADAMTS6*, *ADAMTS9*, and *ADAMTS12* to be a useful tool for distinguishing normal cells from cancer cells. The AUCs from the ROC curves generated for classifying cancer and normal samples ranged from 0.86 to 0.99 for both the LUAD and LUSC subtypes. The methylation level of different CpG sites among selected *ADAMTS* members can reflect patient survival, indicating its potential value as a prognostic marker. It should be noted that both higher and lower methylation of certain CpG sites correlates with shorter NSCLC patient survival, with the region depending on the type of cancer, i.e., LUAD or LUSC. Several publications have linked the expression of *ADAMTS* genes to OS among cancer patients. Lee et al. found that high *ADAMTS8* expression correlated with longer OS for LUAD patients [26]. Elsewhere, lower *ADAMTS9* expression and higher *ADAMTS12* expression were found to be associated with shorter survival among NSCLC patients, particularly those with a LUAD subtype. In contrast, for LUSC patients, a lower expression of *ADAMTS6* was associated with longer survival [27].

This is the first study to describe the influence of CpG site methylation of *ADAMTS* members on the OS of NSCLC patients.

## 4. Materials and Methods

The flowchart comprising the study workflow is presented in Figure 13.

### 4.1. TIMER

The Tumor Immune Estimation Resource (TIMER) database enables comprehensive analyses of immune infiltrates in various types of cancer. It provides a wealth of information about the clinical, immunological, and genomic features of the included tumors (https://cistrome.shinyapps.io/timer/; accessed 21 May 2024) [28].

The expression of three selected genes from the *ADAMTS* family was compared between cancer tissue and normal tissue using the DiffExp module. The data illustrate the differential expression of the gene between different types of cancers from The Cancer Genome Atlas (TCGA) project and compares them with adjacent non-cancerous tissues where such information is available. TCGA is a research network that gathers and analyzes molecular, protein, and epigenetic data of human tumors, which enables, through bioinformatics analyses, the ability to better understand the molecular basis of various diseases, as well as search for new possible methods of treatment or quicker diagnosis. The datasets chosen for this study were as follows: Lung Adenocarcinoma [LUAD] March 2017 and Lung Squamous Cell Carcinoma [LUSC] March 2017. Statistical significance was assessed using the Wilcoxon test, and *p*-values were marked with asterisks: *—*p* < 0.05; **—*p* < 0.01; ***—*p* < 0.001.

### 4.2. UALCAN

UALCAN (The University of Alabama at Birmingham Cancer Data Analysis Portal) is an extensive, easy-to-use, publicly available bioinformatics database that enables multi-level analyses of -omics data for many types of cancer. UALCAN aims to provide easy access to OMICS cancer data and enables new biomarker identification or in silico validation of selected interesting genes. It also provides access to various tools for graphs and chart creation. The portal, using data from the TCGA project, allows for a broad analysis of the expression of specific genes. The tool also gives the opportunity to assess the degree of methylation of the promoter region of selected genes in normal and cancerous tissue. The analysis can be extended with additional assessment criteria, such as sex, age, ethnicity, cancer stage, smoking status, the presence of *TP53* gene mutation, and the number of lymph node metastases. Similarly, the datasets chosen for this study were as follows: ‘Lung Adenocarcinoma [LUAD] March 2017’ and ‘Lung Squamous Cell Carcinoma [LUSC] March 2017’. These comprised 473 LUAD and 370 LUSC patients. The detailed clinicopathological characteristics are presented in Table 1.

The data contained in UALCAN (https://ualcan.path.uab.edu/; accessed 10 May 2024) [29] were used to generate individual box–whisker plots illustrating the level of promoter methylation of three genes of the *ADAMTS* family in patients with LUAD and LUSC compared with healthy controls. The platform also made it possible to assess the degree of gene methylation in accordance with selected clinicopathological factors in these patients. The methylation level of a gene promoter is expressed by the beta value, which ranges from 0 (no methylation) to 1 (full promoter methylation). Statistical significance was assessed using Student’s *t*-test.

### 4.3. MethMarkerDB

MethMarkerDB (https://methmarkerdb.hzau.edu.cn/home; accessed 6 June 2024) [30,31,32] is a database that integrates literature-based gene markers, differential methylation region (DMR) markers, and diagnostic and prognostic evaluation functions. Unsupervised clustering of cancer samples and corresponding normal samples was performed using the R ‘pheatmap’ package (https://CRAN.R-project.org/package=pheatmap; accessed 6 June 2024), which enabled further differential analysis of methylation. The SMART2 algorithm was applied to detect DMRs across the whole genome within various cancer subtypes. The SMART2 tool enabled q-value calculation, with the result reflecting the statistical significance of the DMR in cancer samples compared with normal samples.

### 4.4. MethSurv

MethSurv (https://biit.cs.ut.ee/methsurv/; accessed 4 June 2024) [33] is an interactive web portal that provides univariable and multivariable survival analysis using the TCGA database based on DNA methylation biomarkers. The information about survival was available for 461 LUAD and 372 LUSC patients. Beta values (ranging from 0 to 1), calculated for every CpG site, were used to represent DNA methylation. The calculations were based on the M/(M + U + 100) formula, where M and U are the methylated and unmethylated intensities, respectively.

## 5. Conclusions

In conclusion, the degree of methylation of the promoter regions in genes encoding ADAMTS family proteins could significantly influence the course of lung squamous-cell carcinoma and lung adenocarcinoma. Increased promoter methylation could reduce gene expression, contributing to cancer progression. For *ADAMTS12*, additional epigenetic factors may also play a role. Clinical and pathological factors could impact promoter methylation, necessitating further in-depth studies for each gene.

Our analysis indicates that squamous-cell carcinoma and adenocarcinoma display different profiles of expression regulation and promoter methylation, suggesting that the two NSCLC subtypes possess disparities at the molecular level. Additionally, the expression of *ADAMTS* genes and their specific DMRs may serve as prognostic markers correlating with patient OS. Assessing *ADAMTS* gene methylation could become a diagnostic tool for differentiating NSCLC subtypes and potentially guide therapeutic strategies. Further research is needed to fully understand the activity and mechanisms of ADAMTS family proteins.

## Figures and Tables

**Figure 1 ijms-26-00934-f001:**
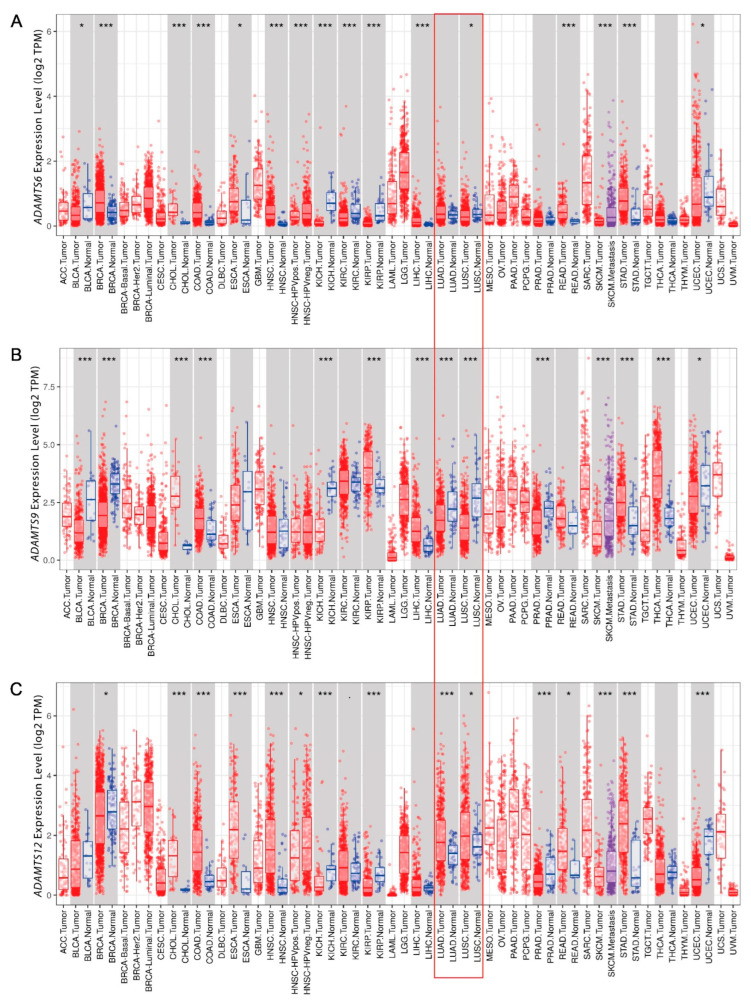
The mRNA expression of *ADAMTS6* (**A**), *ADAMTS9* (**B**), and *ADAMTS12* (**C**) genes in various cancers and adjacent normal tissues from the Tumor Immune Estimation Resource (TIMER) database. Data are shown as red (cancer tissue) and blue (normal tissue) boxplots; raw data are represented by dots. Significant differences are denoted with asterisks: * *p* < 0.05, and *** *p* < 0.001, respectively. Red border signifies lung adenocarcinoma (LUAD) and squamous-cell carcinoma (LUSC) data.

**Figure 2 ijms-26-00934-f002:**
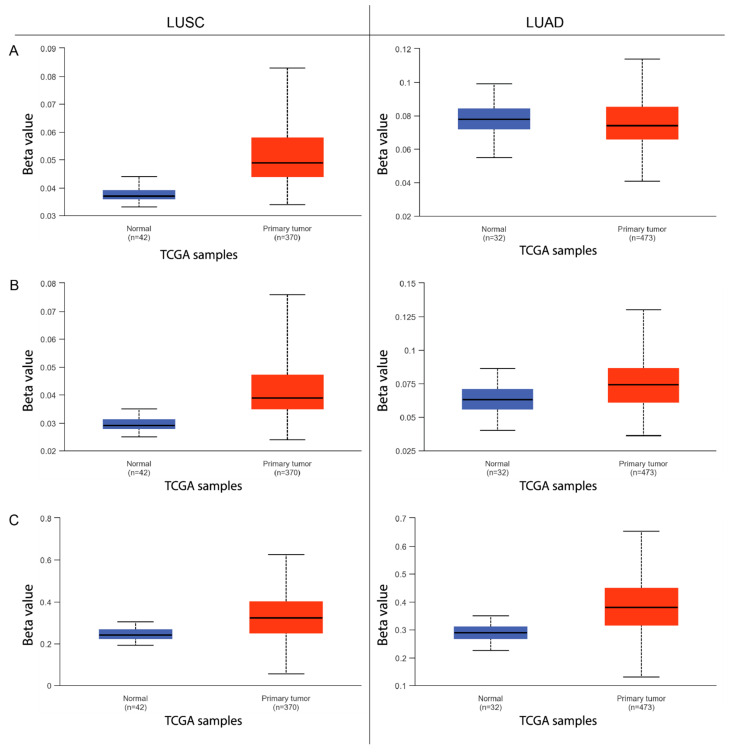
The degree of methylation of the *ADAMTS6* (**A**), *ADAMTS9* (**B**), and *ADAMTS12* (**C**) promoter regions in LUAD and LUSC samples from the UALCAN (The University of Alabama at Birmingham Cancer Data Analysis Portal) database. Data are shown as red (primary tumor) and blue (normal tissue) boxplots. TCGA - The Cancer Genome Atlas.

**Figure 3 ijms-26-00934-f003:**
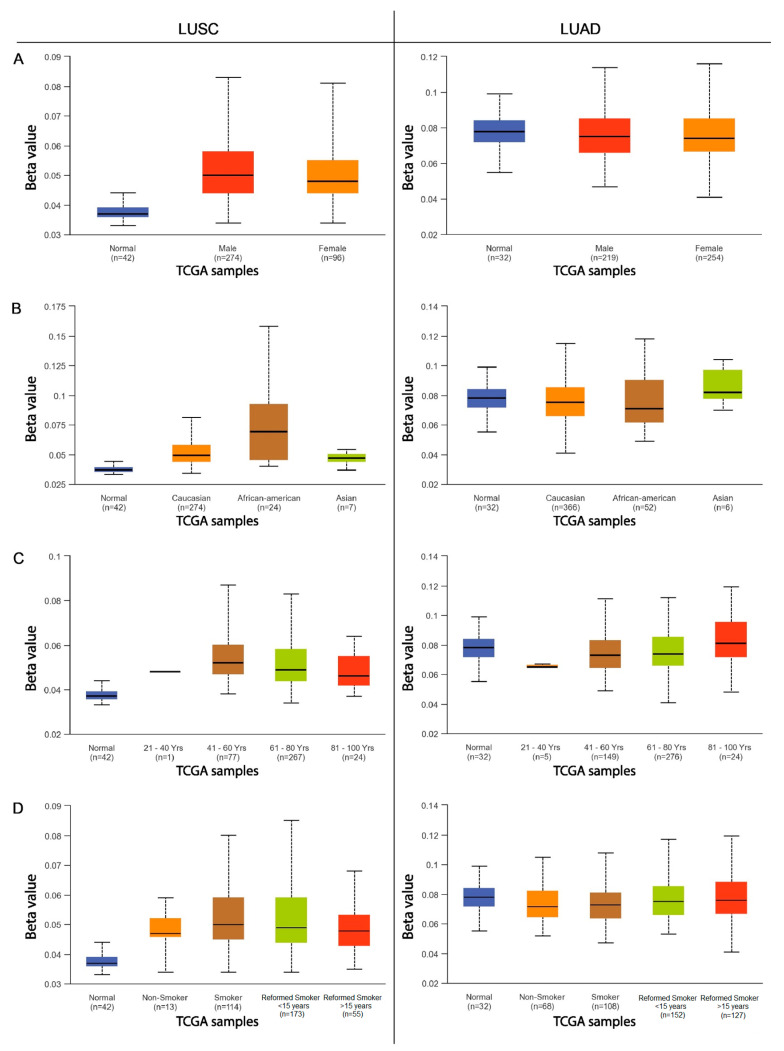
The methylation level of the *ADAMTS6* gene promoter region in relation to various clinicopathological and demographic factors ((**A**)—patient sex; (**B**)—ethnics; (**C**)—age group; (**D**)—smoking status) in the group of patients with LUAD and LUSC.

**Figure 4 ijms-26-00934-f004:**
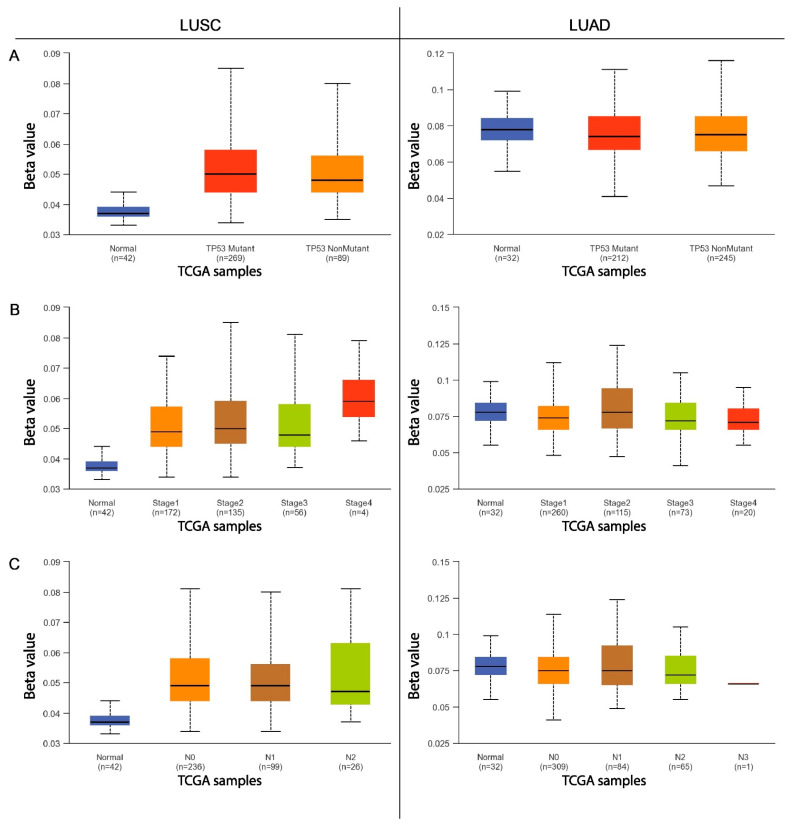
The methylation level of the *ADAMTS6* gene promoter region in relation to various clinicopathological and demographic factors ((**A**)—*TP53* status; (**B**)—disease stage; (**C**)—nodal involvement) in the group of patients with LUAD and LUSC.

**Figure 5 ijms-26-00934-f005:**
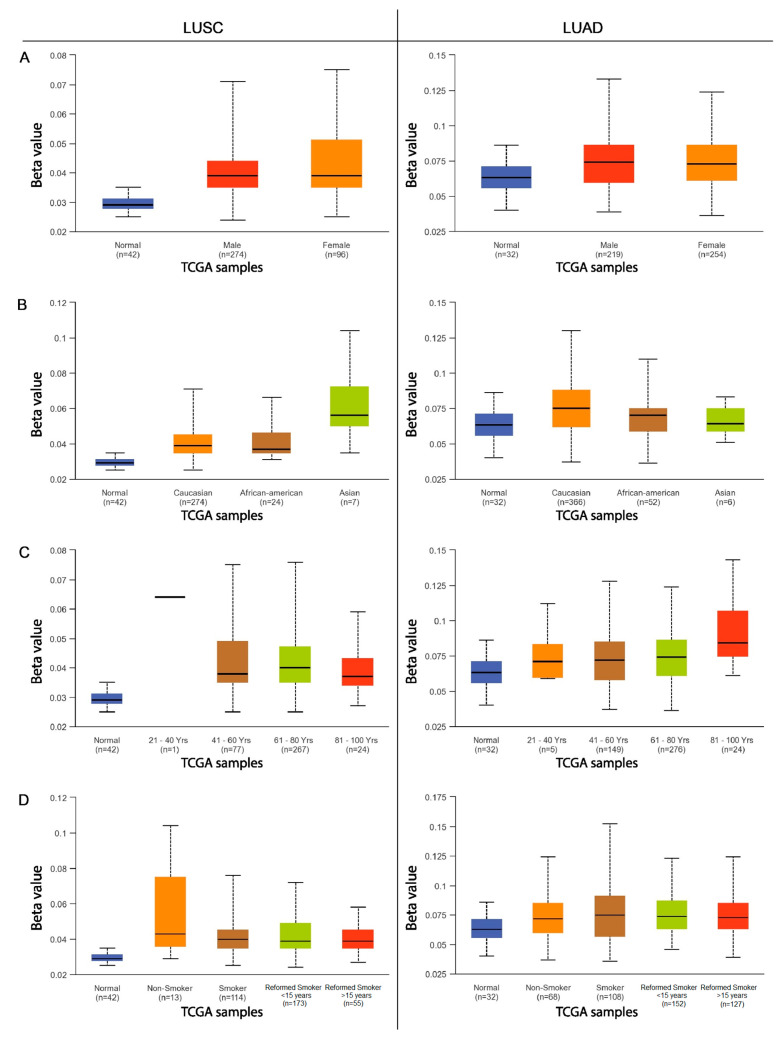
The methylation level of the *ADAMTS9* gene promoter region in relation to various clinicopathological and demographic factors ((**A**)—patient sex; (**B**)—ethnics; (**C**)—age group; (**D**)—smoking status) in the group of patients with LUAD and LUSC.

**Figure 6 ijms-26-00934-f006:**
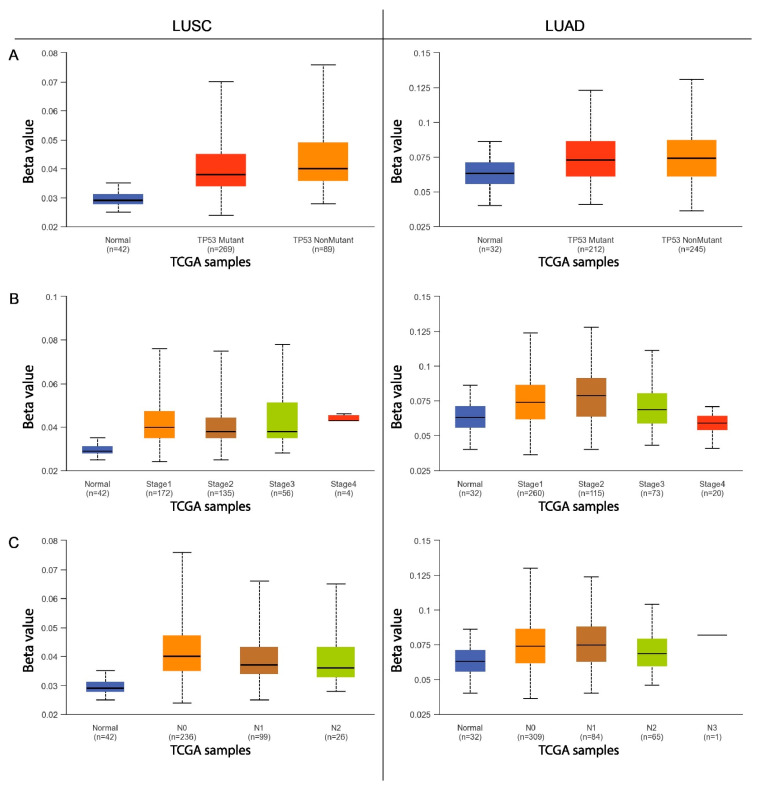
The methylation level of the *ADAMTS9* gene promoter region in relation to various clinicopathological and demographic factors ((**A**)—*TP53* status; (**B**)—disease stage; (**C**)—nodal involvement) in the group of patients with LUAD and LUSC.

**Figure 7 ijms-26-00934-f007:**
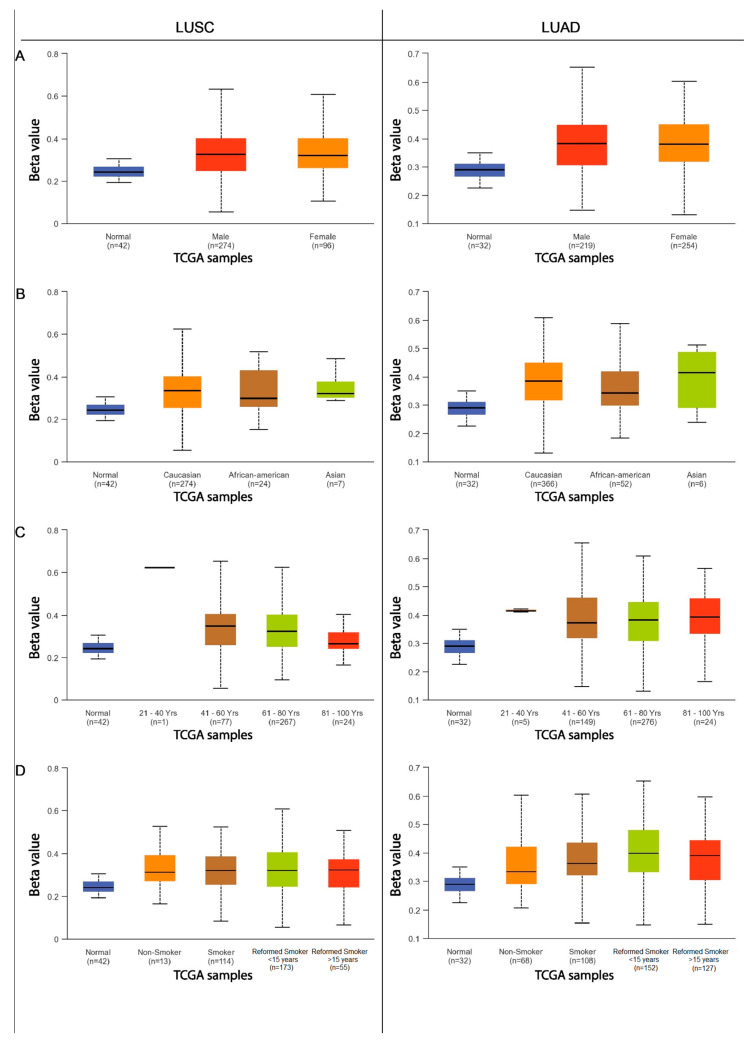
The methylation level of the *ADAMTS12* gene promoter region in relation to various clinicopathological and demographic factors ((**A**)—patient sex; (**B**)—ethnics; (**C**)—age group; (**D**)—smoking status) in the group of patients with LUAD and LUSC.

**Figure 8 ijms-26-00934-f008:**
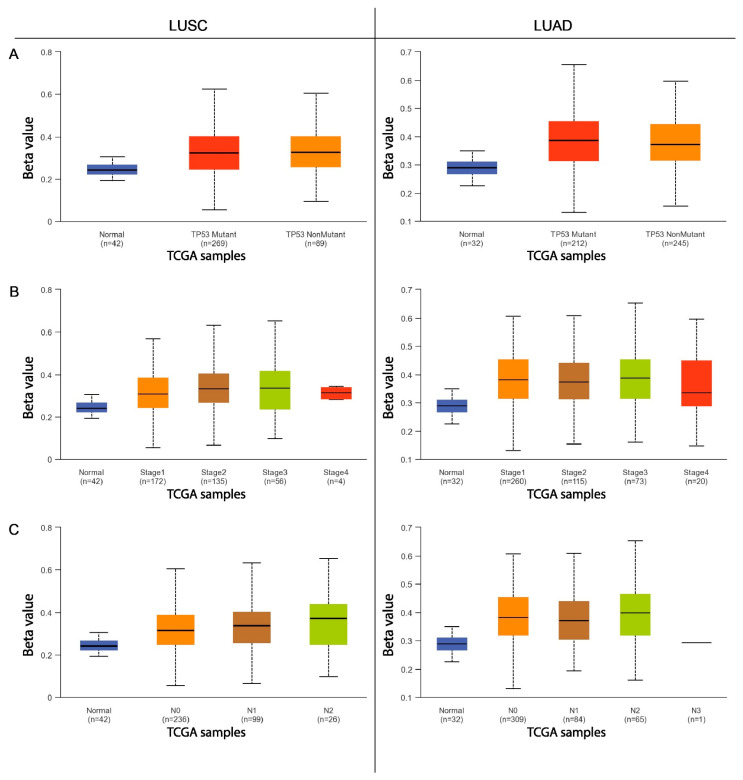
The methylation level of the *ADAMTS12* gene promoter region in relation to various clinicopathological and demographic factors ((**A**)—*TP53* status; (**B**)—disease stage; (**C**)—nodal involvement) in the group of patients with LUAD and LUSC.

**Figure 9 ijms-26-00934-f009:**
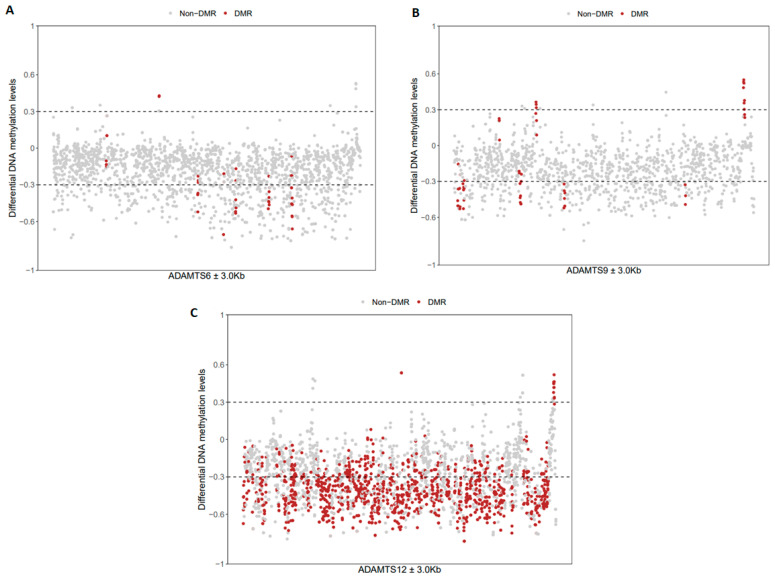
Differentially methylated regions (DMRs) in the *ADAMTS6* (**A**), *ADAMTS9* (**B**), and *ADAMTS12* (**C**) genes in the lung cancer cohort. The grey dots represent Non-DMR, while red dots stand for DMRs spanning *ADAMTS* genes.

**Figure 10 ijms-26-00934-f010:**
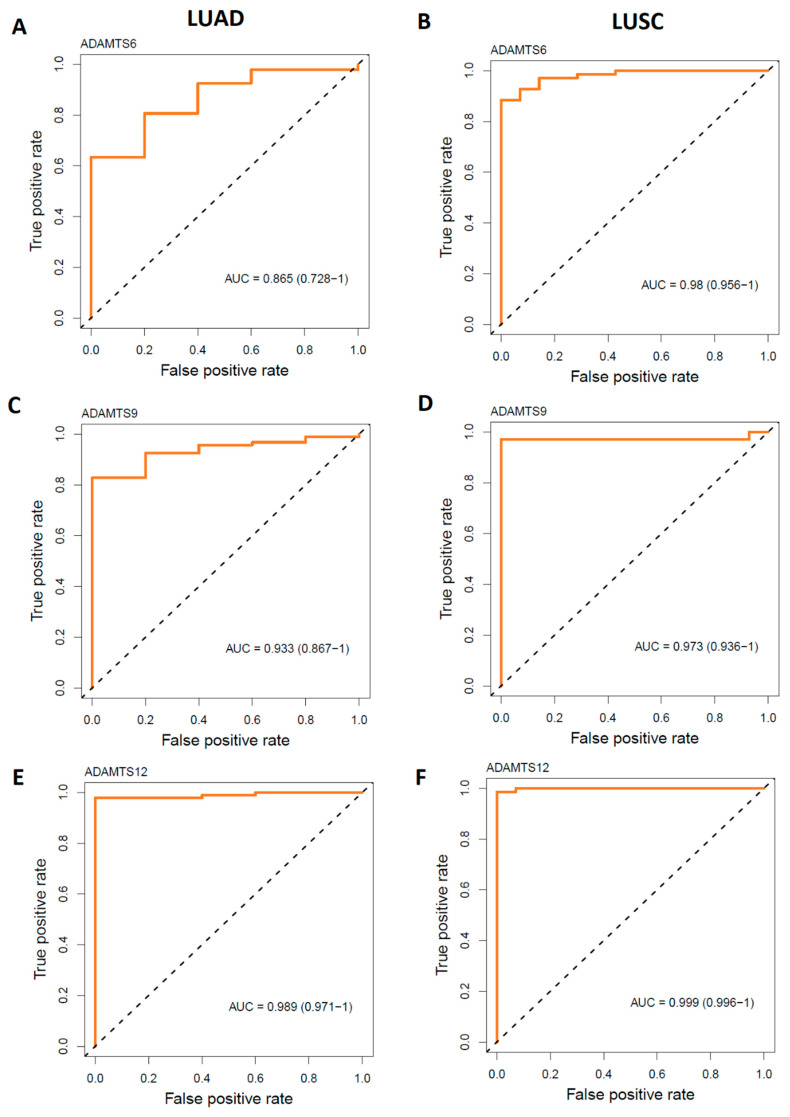
ROC (receiver operating characteristic) curves for DMRs overlapped with *ADAMTS6*, *ADAMTS9*, and *ADAMTS12* (±3kb) genes in classifying cancer and normal samples in the TCGA LUSC (**B**,**D**,**F**) and LUAD (**A**,**C**,**E**) cohort.

**Figure 11 ijms-26-00934-f011:**
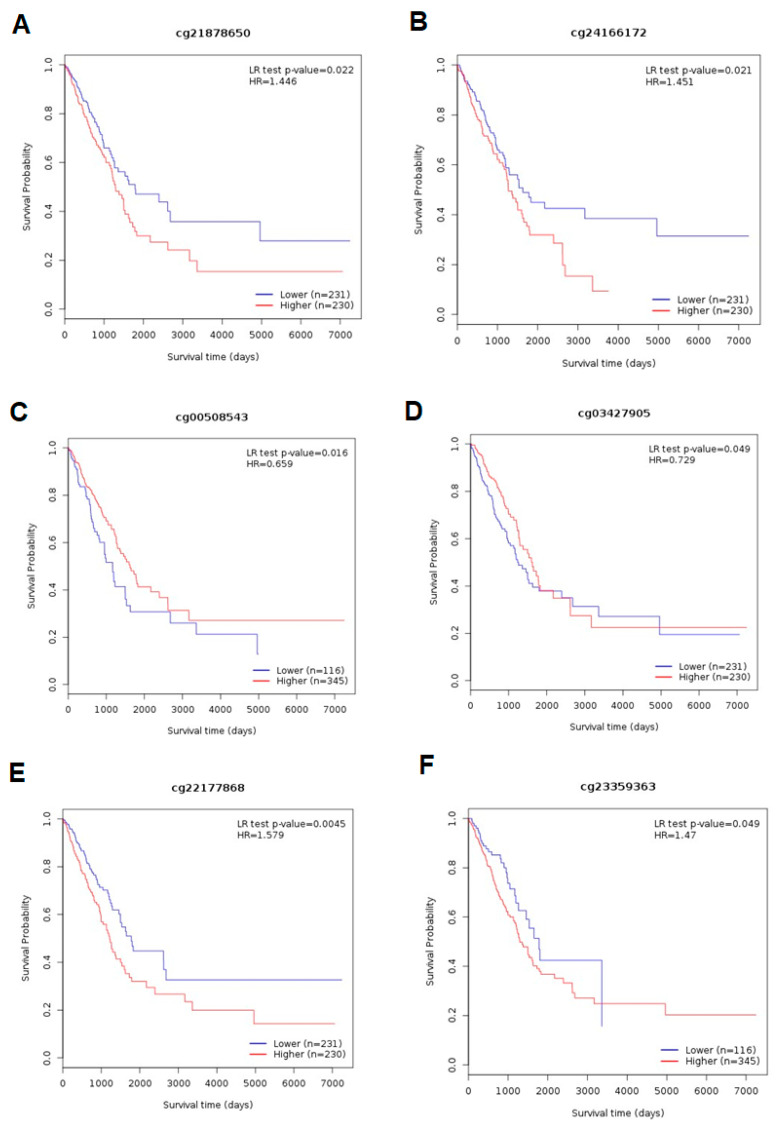
MethSurv database analysis of the effect of *ADAMTS6* (**A**,**B**), *ADAMTS9* (**C**–**E**), and *ADAMTS12* (**F**) methylation levels on the prognosis of LUAD. A significant relationship was observed between patient OS and promoter methylation in LUAD (cg21878650, cg24166172, cg00508543, cg03427905, cg22177868, and cg23359363; all *p* < 0.05).

**Figure 12 ijms-26-00934-f012:**
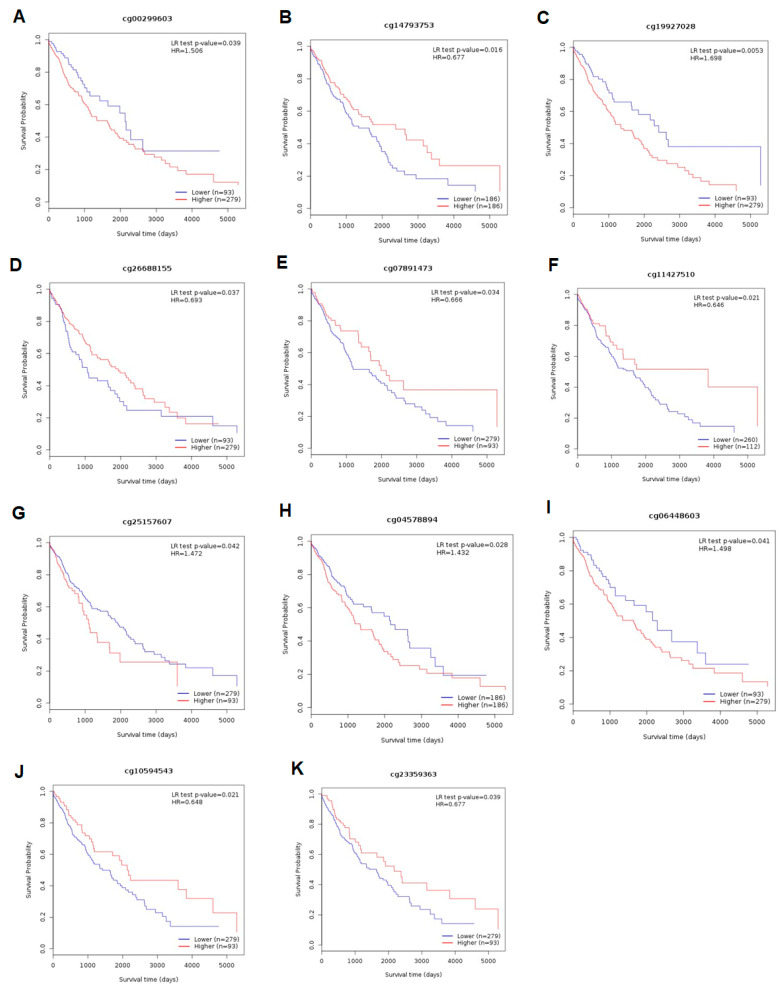
MethSurv database analysis of the effect of *ADAMTS6* (**A**–**D**), *ADAMTS9* (**E**–**G**), and *ADAMTS12* (**H**–**K**) methylation levels on the prognosis of LUSC. A significant relationship was observed between patient OS and promoter methylation in LUSC (cg00299603, cg14793753, cg19927028, cg26688155, cg07891473, cg11427510, cg25157607, cg04578894, cg06448603, cg10594543, and cg23359363; all *p* < 0.05).

**Figure 13 ijms-26-00934-f013:**
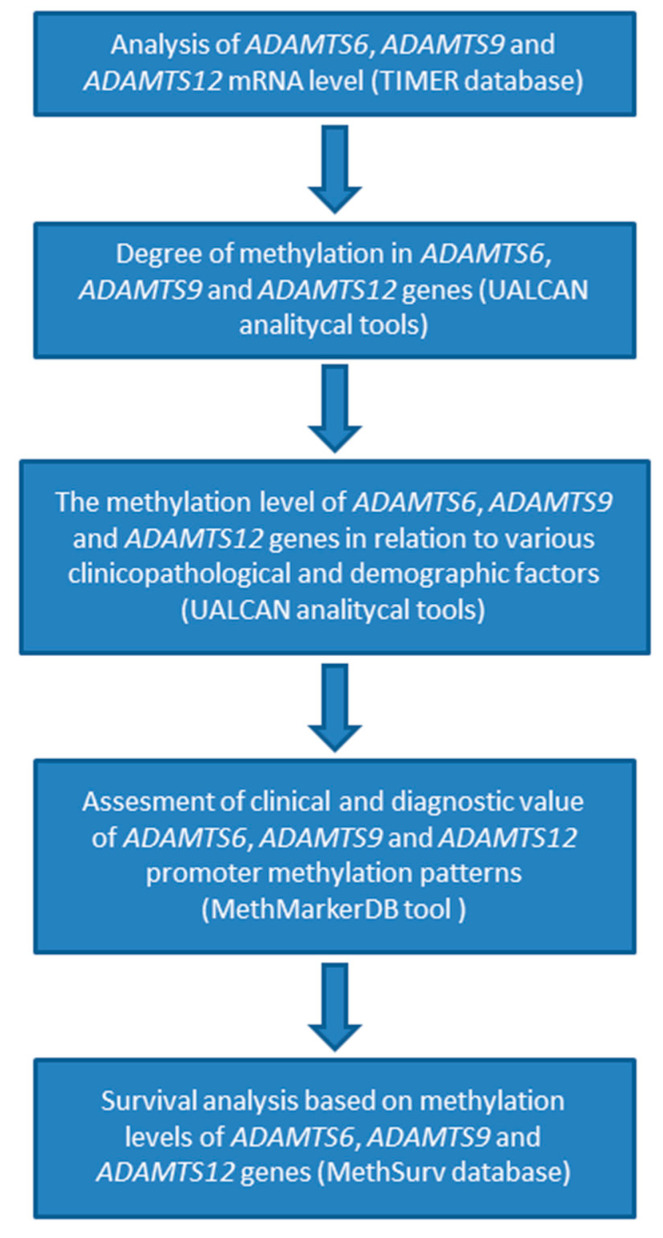
Study design flowchart.

**Table 1 ijms-26-00934-t001:** Clinical characteristics of the TCGA cohort taken for analysis.

Clinical Characteristic	Number of Patients LUAD	Number of Patients LUSC
Sex		
Men	219	274
Women	254	96
Age group		
21–40 years	5	1
41–60 years	149	77
61–80 years	276	267
81–100 years	24	24
Smoking status		
Non-smoker	68	13
Smoker	108	114
Reformed smoker (<15 years)	152	173
Reformed smoker (>15 years)	127	55
*TP53* status		
Mutant	212	269
Non-mutant	245	89
Disease stage		
Stage 1	260	172
Stage 2	115	135
Stage 3	73	56
Stage 4	20	4
Nodal involvement		
N0	309	236
N1	84	99
N2	65	26
N3	1	0
Ethnics		
Caucasian	366	274
African American	52	24
Asian	6	7

## Data Availability

The datasets used in this study are available publicly (links to the archives: https://cistrome.shinyapps.io/timer/ (accessed on 21 May 2024); https://ualcan.path.uab.edu/ (accessed on 10 May 2024); https://methmarkerdb.hzau.edu.cn/home (accessed on 6 June 2024); https://CRAN.R-project.org/package=pheatmap; accessed on 6 June 2024, https://biit.cs.ut.ee/methsurv/ (accessed on 4 June 2024)).

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
