# Peer review of "Assessment of Methylation in Selected ADAMTS Family Genes in Non-Small-Cell Lung Cancer"

_ijms, 2025, doi:10.3390/ijms26030934_

Round 1

Reviewer 1 Report

Comments and Suggestions for Authors

In the manuscript, the authors examined the methylation assessment of selected ADAMTS family genes in non-small cell lung cancer. This is a significant and intriguing health issue. I have a few comments and suggestions:

1. The strength of the study is the research design, descriptions of methods and results presentation.

2. Moreover, the introduction and discussion are interesting and well-written. I enjoyed reading this article.

3. Line 26, 27, 439, 440 and 442: I recommended changing “may” to “could”.

4. Lines 57-58: I suggest adding more citations if the authors indicate that there was more than one study.

5. If the study aimed to analyze gene methylation in NSCLC subtypes, I do not understand why information about other types of cancer is included in the results section. Such information would be better suited for the discussion section.

6. Including a flowchart could further enhance the article (materials and methods section).

7. I feel there is a lack of explanation for the TCGA and UALCAN.

Reviewer 2 Report

Comments and Suggestions for Authors

This article evaluates the impact of methylation in selected ADAMTS family genes 2 in non-small cell lung cancer. This is very interesting topic and very interesting results are presented.

But the manuscript is a little bit confusing. In the introduction it was written that the aim of the study was to evaluate the degree of methylation of the 73 ADAMTS6, ADAMTS9, and ADAMTS12 genes in NSCLC. Than in the section 2 Results, the data about these genes in other cancers are also presented. The role of ADAMTS6 and ADAMTS12 is evaluated, but need to be enriched, but the role of ADAMTS9 is not. Further, the role of TP53 was also evaluated but there are no data about the impact of TP53 itself. The introduction and results must be improved.

The section 4 Materials and Methods must be improved. The databases are described, but there are no data about the patients which tissues have been used, i.e. stage of disease, the tissues used in the analysis were obtained before the treatment or after the treatment was introduced. In a group of patients with adenocarcinoma were there patients with driver mutations? If yes, did it have impact on results? The section 4 must be enriched with the data about the patients whose tissues were used. Maybe authors could also make a table with the patient's data in the section results.

Also it was found that higher or lower methylation at the CpG sites was associated with survival. But there are no data about the applied treatment and stage of disease. Nowadays with introduction of molecular and immunotherapy the outcomes of patients with NSCLC are significantly improved. The authors should include the data about applied therapies and stage of disease and to evaluate the impact on the results.

Round 2

Reviewer 2 Report

Comments and Suggestions for Authors

No further comments.